# Genome-Wide Identification and Expression Analysis of *NAC* Gene Family Members in Seashore Paspalum Under Salt Stress

**DOI:** 10.3390/plants13243595

**Published:** 2024-12-23

**Authors:** Xuanyang Wu, Xiaochen Hu, Qinyan Bao, Qi Sun, Pan Yu, Junxiang Qi, Zixuan Zhang, Chunrong Luo, Yuzhu Wang, Wenjie Lu, Xueli Wu

**Affiliations:** 1College of Grassland Science, Qingdao Agricultural University, Qingdao 266109, China; 15216553920@163.com (X.W.); wrnfw@163.com (X.H.); baoqy17@lzu.edu.cn (Q.B.); 15124895411@163.com (Q.S.); 13332190227@163.com (P.Y.); qjx1658936266@163.com (J.Q.); 13903462202@163.com (Z.Z.); lcr15286377820@163.com (C.L.); wang_yuzhu2023@163.com (Y.W.); m17606429183@163.com (W.L.); 2College of Pastoral Agriculture Science and Technology, Lanzhou University, Lanzhou 730050, China; 3Shandong Key Laboratory for Germplasm Innovation of Saline-alkaline Tolerant Grasses and Trees, Qingdao Agricultural University, Qingdao 266109, China

**Keywords:** *Paspalum Vaginatum*, *NAC* TFs, salt stress response, molecular breeding, qRT-PCR

## Abstract

The *NAC* gene family plays a crucial role in plant growth, development, and responses to biotic and abiotic stresses. *Paspalum Vaginatum*, a warm-season turfgrass with exceptional salt tolerance, can be irrigated with seawater. However, the *NAC* gene family in seashore paspalum remains poorly understood. In this study, genome-wide screening and identification were conducted based on the *NAC* (*NAM*) domain hidden Markov model in seashore paspalum, resulting in the identification of 168 *PvNAC* genes. A phylogenetic tree was constructed, and the genes were classified into 18 groups according to their topological structure. The physicochemical properties of the *PvNAC* gene family proteins, their conserved motifs and structural domains, cis-acting elements, intraspecific collinearity analysis, GO annotation analysis, and protein–protein interaction networks were analyzed. The results indicated that the majority of PvNAC proteins are hydrophilic and predominantly localized in the nucleus. The promoter regions of *PvNACs* are primarily enriched with light-responsive elements, *ABRE* motifs, *MYB* motifs, and others. Intraspecific collinearity analysis suggests that *PvNACs* may have experienced a large-scale gene duplication event. GO annotation indicated that *PvNAC* genes were essential for transcriptional regulation, organ development, and responses to environmental stimuli. Furthermore, the protein interaction network predicted that *PvNAC73* interacts with proteins such as *BZIP8* and *DREB2A* to form a major regulatory hub. The transcriptomic analysis investigates the expression patterns of *NAC* genes in both leaves and roots under varying durations of salt stress. The expression levels of 8 *PvNACs* in roots and leaves under salt stress were examined and increased to varying degrees under salt stress. The qRT-PCR results demonstrated that the expression levels of the selected genes were consistent with the FPKM value trends observed in the RNA-seq data. This study established a theoretical basis for understanding the molecular functions and regulatory mechanisms of the *NAC* gene family in seashore paspalum under salt stress.

## 1. Introduction

Plant transcription factors (TFs) can bind to specific DNA motifs to regulate expression [1], playing crucial roles in plant growth, development, and responses to abiotic and biotic stresses [2]. Considering the significance of TFs in responding to a variety of external stimuli, the analysis of their functional mechanisms has become essential for aspects such as plant improvement. As genomic sequencing and transgenic technologies have advanced, a large number of TFs have been identified at the molecular level and in terms of their functionality, such as those in the *WRKY* [3,4], *bHLH* [5], *NAC* [6] families, and more.

The typical *NAC* protein comprises a conserved *NAC* domain and a highly variable transcriptional activation region (TAR) [7]. The protein possesses a highly conserved N-terminal *NAC* domain and a variety of C-terminal structural domains, which have been extensively studied [8]. The N-terminal *NAC* domain, comprising approximately 160 amino acids, is capable of binding to the cis-elements of its target genes. In contrast, the C-terminal domain serves as a critical region for interactions with other proteins, thereby playing a pivotal role in the regulation of gene expression [9,10]. Due to the presence of the conserved *NAC* domain, *NAC* genes are relatively easy to screen, and research on the *NAC* gene family has emerged rapidly in recent years [11,12].

*NAC* transcription factors play a pivotal role in plant growth, development, and responses to abiotic stresses by regulating various signaling networks, modulating plant hormone levels, and influencing miRNA activity. They are involved in responding to poor construction such as drought [13], salt stress [14], light exposure [15], and temperature fluctuations (e.g., low [16] and high temperatures [17]). These functions highlight the essential role of *NAC* TFs in enhancing plant resilience under diverse stress conditions. The *NAC* genes are extensively distributed across plant species and represent one of the largest families of transcription factors. They have been identified in various species, including *Arabidopsis* [18], rice (*Oryza sativa* L.) [19], and sorghum (*Sorghum bicolor*) [20]. Research on the interaction between plant hormones and *NAC* TFs is also extensive, such as studies on the interplay between *NAC* TFs and the ethylene and ABA pathways in controlling the development and ripening of tomato (*Solanum lycopersicum* L.) fruits [21]; *StNAC53* [22], *OsNAC52* [23], and *OsNAC45* [24] have been studied for their ability to enhance ABA sensitivity and salt tolerance in transgenic plants. For example, the overexpression of microRNAs targeting *NAC* transcription factors improves the drought and salt tolerance of rice through an ABA-mediated pathway [25]. The conserved *miR164* (*OMTN1*–*OMTN6*) targets *NAC* genes, negatively regulating drought resistance in rice [26]. The development of lateral roots in potatoes (*Solanum tuberosum* L.) is regulated by the *NAC* transcription factor Stu-mi164, which promotes the formation of lateral roots [27] and others. These related studies indicate that *NAC* TFs play an essential role in the growth and development of plants.

Seashore paspalum (*Paspalum Vaginatum* O. Swartz, 2n = 2x = 20) is a halophytic warm-season grass, which is commonly used on landscape areas, golf courses, and athletic fields [28,29]. It is widely distributed around the globe, particularly in coastal areas of tropical and subtropical regions. This plant is renowned for its vibrant color, fine turf texture, vigorous growth, and exceptional adaptability and stress resistance. It can adapt to seawater irrigation, making it an ideal choice for greening projects in saline-alkali areas and coastal regions of tropical and subtropical zones [30,31]. Seashore paspalum exhibits significant potential for ecological restoration and the improvement of saline-alkali lands due to its exceptional salt tolerance and resilience [32]. As a halophytic plant, seashore paspalum exhibits superior ion regulation capabilities, characterized by its ability to maintain high concentrations of K^+^ in shoots while limiting the transfer of Na^+^ from roots to shoots, thereby sustaining a favorable K^+^/Na^+^ ratio. This mechanism allows the plant to mitigate ion-specific damage induced by salt stress [33,34]. Additionally, seashore paspalum enhances its Ca^2+^ signaling transduction pathway in response to Na^+^ accumulation, supporting the activity of major antioxidant enzymes, and further bolstering its resilience to salt stress [35]. The *Agrobacterium*-mediated transformation of seashore paspalum has been successfully established [36]. The complete genome assembly for the seashore paspalum genotype (PI 509022) has been completed [37], along with the provision of a transcriptome profile for salt resistance [35]. Selection of transgenic plants expressing *CdNF-YC* genes has demonstrated enhanced drought and salt tolerance [34], with additional gene functions related to salt tolerance being identified [38]. The genomic data and genetic transformation system have provided a foundational basis for exploring and validating salt tolerance genes in seashore paspalum. However, the role of the *NAC* gene family in the identification of salt-tolerant genes in seashore paspalum has not yet been given focused attention.

The *NAC* gene family is crucial for salt tolerance [39,40]; however, to date, no genome-wide identification and analysis of the *NAC* gene family have been reported in seashore paspalum. This study aims to fill this knowledge gap by conducting a comprehensive genome-wide analysis of the *NAC* gene family in seashore paspalum and investigating their expression patterns under both short-term and long-term salt stress conditions. Specifically, we identified the *NAC* gene family and analyzed their differential expression in leaves and roots under both short-term and long-term salt stress using transcriptomic analysis in seashore paspalum. This study thoroughly investigates the *PvNAC* gene family in seashore paspalum, covering aspects such as evolutionary relationships, gene structure, motifs, cis-regulatory elements, GO annotation analysis, and protein–protein interactions. The transcriptomic analysis explores *NAC* gene expression patterns in leaves and roots under both short-term and long-term salt stress. These findings establish a theoretical basis for understanding the molecular functions and regulatory mechanisms of the *NAC* gene family in seashore paspalum under salt stress, aiding in the development of new germplasm and identification of salt-tolerant genes.

## 2. Results

### 2.1. Identification and Protein Characterization of the PvNAC Gene Family

Using HMM profiles to filter redundant sequences for both *NAC* and *NAM* domains, a total of 168 *PvNAC* genes were identified and sequentially named from *PvNAC01* to *PvNAC168* according to their chromosomal location (Appendix A). Phylogenetic analysis grouped these genes into 18 subgroups, consistent with the classification of NAC gene families in *Arabidopsis* and rice (Appendix A) [18]. Compared to the number of *NAC* TFs in other species, such as rice with 151 members [19], sorghum with 183 members [20], sugarcane (*Saccharum officinarum*) with 85 members [41], grape (*Vitis vinifera* L.) with 75 members [42], cocoa (*Theobroma cacao* L.) with 102 members [43], mulberry (*Morus alba* L.) with 79 members [44], pepper (*Capsicum annuum* L.) with 104 members [45], and canola (*Brassica napus*) with 60 members [46], *PvNACs* is considered a species with a larger family size.

Physicochemical analysis (Appendix A) shows that most PvNAC proteins (91.66%) have lengths between 200 and 600 amino acid residues, while only a small proportion (2.97%) exceed 700 residues. The average sequence length of PvNAC proteins is 392 amino acid residues, ranging from a minimum of 134 residues (PvNAC36) to a maximum of 1403 residues (PvNAC39). Protein lengths range from 134 residues (PvNAC36) to 1403 residues (PvNAC39), with corresponding molecular weights from 14,673 to 157,368 Da and an average of 39,827.3 Da. The isoelectric points (pI) range from 3.17 to 10.40. The instability coefficients vary between 30.63 and 77.05, resulting in the majority of PvNAC proteins being classified as unstable, with only 26 PvNAC proteins having an instability coefficient below 40, which are considered stable proteins. In terms of hydrophilicity, all 168 PvNAC proteins exhibit negative hydrophilicity indices, indicating their hydrophilic nature. Subcellular localization analysis predicts that PvNAC proteins are distributed across various compartments, including chloroplasts, cytoplasm, mitochondria, and the plasma membrane, with most members localized in the nucleus.

### 2.2. Classification and Phylogenetic Relationships of PvNACs

To understand the evolutionary relationships among the 168 motifs of the *PvNAC* gene family, we constructed a phylogenetic tree using MEGA11. Based on the results of the phylogenetic tree (Figure 1; Appendix A), 18 NAC subfamilies have already been identified in *Arabidopsis* and rice [18]. In this study, based on the topological structure of the *PvNAC* phylogenetic tree, we redefined 18 subfamilies. Among them, the PvNAC12 subgroup has the highest number of genes with 24, while the *PvNAC06* subgroup has the fewest with only 3. It is worth noting that PvNAC127, PvNAC20, PvNAC125, PvNAC43, PvNAC163, and PvNAC166 do not have a clear classification on the evolutionary tree. It is speculated that these genes may undergo specific mutations or perform certain specialized functions during the process of species evolution [47].

After constructing a phylogenetic tree of PvNAC proteins alongside AtNAC proteins (Appendix A), we found that PvNAC86, PvNAC114, and PvNAC151 clustered closely with the Arabidopsis AtNAC06 (ATAF1) and AtNAC26. In Arabidopsis, AtNAC06 (ATAF1) plays a crucial role in the ABA response pathway and salt stress tolerance by regulating reactive oxygen species (ROS) scavenging mechanisms, thereby enhancing plant resistance to salt stress [48]. This suggests that they may play similar roles in responding to salt stress.

### 2.3. Motifs and Gene Structure of PvNAC

To explore the structural and evolutionary features of the *PvNAC* gene family, we used Tbtools to visualize protein motifs, including introns, exons, and conserved sequences [49]. Motifs 1 to 15, identified as the most prevalent structural elements within the PvNAC gene family, are depicted in Appendix A and detailed in Appendix A. Notably, *PvNAC01*, *PvNAC10*, *PvNAC17*, *PvNAC23*, *PvNAC163*, and *PvNAC106* lack these motifs but still retain the conserved NAC domain. Motif 9 was exclusively identified in specific subfamilies. The exon-intron structure analysis highlights evolutionary diversity within the family, with most members having two exons, while a few exhibit either more or fewer than two exons. Regarding protein domain structure, there are instances where a single gene harbors features of different protein domains (Appendix A). For example, the PvNAC39 protein contains features of the *NAM* superfamily, *WRKY*, and *NB-ARC* superfamily domains, suggesting that its multifunctionality contributes to this diversity.

### 2.4. Cis-Element Analysis of the PvNAC Gene Family

To investigate the regulatory relationships between *PvNACs* and other transcription factors, we extracted a 2000 bp promoter region of the *PvNAC* genes and analyzed their cis-acting elements using PlantCARE, followed by data visualization with Tbtools. We identified various functional elements, with almost all *NAC* genes possessing “light-responsive” elements, indicating that the *PvNAC* gene family is largely regulated by light exposure. This underscores that light is a fundamental factor for the survival of the seashore paspalum species. Additionally, hormone-responsive elements, such as abscisic acid-responsive elements (ABRE), MYB binding sites involved in drought and salt stress responses, gibberellin-responsive elements, salicylic acid-responsive elements, and auxin-responsive elements, were widely present in the *NAC* gene promoters (Appendix A). The presence of ABRE elements indicates that *PvNAC* genes are likely regulated by abscisic acid (ABA), a key hormone in plant stress responses, particularly under salt stress [50]. Similarly, MYB binding sites suggest regulation by MYB transcription factors, which play critical roles in abiotic stress responses, including salt tolerance [51]. This implies that these elements are extensively involved in the transcriptional regulation of PvNAC genes under salt stress. Defense and stress response elements, which are critical in plant responses to abiotic and biotic stresses, were also widely distributed in *PvNAC* gene promoters. For instance, dehydration-responsive elements (DREs) and low-temperature-responsive (LTR) elements were commonly identified in PvNAC promoters, indicating their potential roles in diverse stress signaling pathways. Based on the presence of these cis-elements, we hypothesize that *PvNAC* genes may be involved in ABA-mediated stress response pathways and interact with MYB transcription factors to regulate gene expression under salt stress. The ABRE elements could allow PvNAC promoters to be activated by ABA signaling, leading to the expression of stress-responsive genes. Similarly, MYB binding sites may enable interactions with MYB proteins that modulate stress tolerance mechanisms, such as osmolyte accumulation and reactive oxygen species (ROS) scavenging.

Notably, *PvNAC28* to *PvNAC35* share identical cis-acting elements in their promoters, with highly similar sequences in the promoter regions. This suggests that these *PvNAC* genes likely form a gene cluster or multigene family, and the proteins encoded by these *PvNAC* genes may perform similar functions [52]. This phenomenon also illustrates the unified relationship between gene structure and function.

### 2.5. Collinearity and Evolution Analysis of PvNACs

We conducted an intraspecific collinearity analysis of the *PvNAC* genes using Tbtools to explore homologous relationships among *PvNAC* genes across different chromosomes. The results are as follows (Figure 2). The 168 *PvNAC* genes were mapped based on their chromosomal distribution within the seashore paspalum genome, with duplication events highlighted by orange lines. We found that gene duplication events were distributed across all ten chromosomes, although the distribution was uneven. Moreover, no correlation was found between chromosome length, *NAC* gene count, and collinearity patterns. According to Holub’s definition, a chromosomal region that contains two or more genes within a 200 kb span is considered a tandem duplication event [53]. Upon analysis of the figure, it is evident that several pairs of *NAC* genes constitute tandem duplication event regions in the seashore paspalum genome, notably on chromosomes Chr04, Chr06, Chr03, and Chr09. These findings suggest that some *PvNAC* genes may have originated through chromosomal duplication events. This speculation also explains to some extent why genes like *PvNAC28*, *PvNAC29*, *PvNAC30*, and *PvNAC31*, *PvNAC32*, *PvNAC33*, *PvNAC34*, *PvNAC35*, etc., possess the same promoter sequences.

### 2.6. Functional Annotation Evaluation of PvNAC Genes

To elucidate the functional roles of PvNAC proteins, we performed gene ontology (GO) annotation and enrichment analysis, focusing on the categories of biological process (BP), molecular function (MF), and cellular component (CC). Transcriptome-based GO analysis showed significant enrichment of differentially expressed genes (DEGs) in the MF category for L6 vs. R6 (Figure 3A), L48 vs. R48 (Figure 3B), and L120 vs. R120 (Figure 3C). Conversely, GO enrichment of DEGs specific to the NAC family was predominantly observed in the BP category. The results indicated that PvNAC proteins were significantly enriched in key BP and MF pathways, particularly those related to transcriptional regulation and development (Figure 3D; Appendix A).

Biological process enrichment analysis revealed that these proteins were primarily engaged in the regulation of cellular metabolic processes, positive regulation of transcription (GO:0045893), and responses to various biotic and abiotic stimuli. Notably, PvNAC proteins were highly enriched in organ development processes, including leaf senescence (GO:0010150) and root development (GO:0048364), emphasizing their crucial roles in plant development and stress responses. MF annotation revealed a strong association of PvNAC proteins with DNA-binding transcription factor activity (GO:0003700), sequence-specific DNA binding (GO:0043565), and transcription regulatory region binding (GO:0001067). CC analysis confirmed that PvNAC proteins were predominantly localized to the nucleus (GO:0005634) and membrane-bound organelles (GO:0043231), consistent with their roles as transcription factors. In summary, GO annotations across CC, BP, and MF categories highlighted the pivotal roles of PvNAC proteins in transcriptional regulation, organ development, and environmental stress responses.

### 2.7. Expression Patterns of PvNAC Genes Under Salt Stress

Using RNA-seq data from seashore paspalum exposed to varying levels of salt stress, we comprehensively analyzed the expression patterns of the *PvNAC* gene family in roots and leaves. After transforming the FPKM values, we observed significant changes in the expression levels of *PvNAC* gene family members in response to increasing salt stress. To evaluate gene expression correlations between samples, Pearson correlation coefficients were calculated and visualized as a heatmap. The average Pearson correlation within each group exceeded 0.8, indicating high repeatability among the three replicates within each group (Figure 4A). FPKM was used to normalize for sequencing depth and gene length. After calculating the expression values (FPKM) for all genes in each sample, a box plot was generated to show the distribution of gene expression levels across samples (Figure 4B). Notably, the same genes exhibited distinct expression levels in roots and leaves, potentially reflecting their specialized roles in responding to salt stress. Additionally, *NAC* genes within the same phylogenetic branch tended to exhibit similar expression patterns, suggesting functional correlations or synergistic interactions.

These findings allowed us to predict the potential functions of *PvNAC* genes in plant salt tolerance. For instance, *PvNAC06*, *PvNAC38*, *PvNAC71*, and *PvNAC162* exhibited high root-specific expression, suggesting their critical roles in root salt stress response and tolerance. Similarly, *PvNAC17*, *PvNAC53*, and *PvNAC73* were specifically expressed in leaves, indicating their key roles in leaf salt stress response. More interestingly, genes such as *PvNAC06*, *PvNAC17*, *PvNAC53*, *PvNAC73*, *PvNAC142*, and *PvNAC162* exhibited high expression levels in both leaves and roots, which may indicate that they play a central role in the overall salt tolerance mechanism of the plant (Figure 4C). These analysis results can lay the foundation for studying the molecular mechanisms of salt tolerance function in seashore paspalum.

### 2.8. Validation of Transcriptomic Data Using qRT-qPCR

Using RNA-seq data, we selected eight genes—*PvNAC06*, *PvNAC17*, *PvNAC38*, *PvNAC53*, *PvNAC71*, *PvNAC73*, *PvNAC142*, and *PvNAC162*—for qRT-PCR validation. qRT-PCR analysis confirmed the expression patterns of PvNAC genes in seashore paspalum leaves and roots under salt stress at 0 h, 6 h, 48 h, and 120 h. The results (Figure 5) showed that compared to the 0 h time point, *PvNAC06* expression increased in both leaves and roots under salt stress, with the highest expression observed in roots at 120 h. *PvNAC17* similarly exhibited an increasing trend, with peak expression at 120 h in leaves. *PvNAC38* and *PvNAC71* showed an initial decrease in expression followed by an increase, with maximum levels observed in roots at 120 h. Similarly, *PvNAC53*, *PvNAC73*, and *PvNAC162* demonstrated an initial increase in expression, peaking at 120 h in roots, and followed by a slight decrease. These findings suggest that these genes are regulated in a time-dependent manner under salt stress, indicating their critical roles in specific stages of the salt stress response. Moreover, except for the data for *PvNAC162* in leaves, the expression trends observed through qRT-PCR were consistent with the results obtained from RNA-seq data.

### 2.9. Protein Interaction Network Prediction for PvNACs

To investigate the molecular mechanisms and interaction patterns of *PvNAC* transcription factor family members in regulating salt tolerance, we constructed a protein–protein interaction (PPI) network based on *Arabidopsis* homologs for the four *PvNAC* genes (*PvNAC06*, *PvNAC17*, *PvNAC38*, *PvNAC73*) that showed significant upregulation in qRT-PCR analysis. The results revealed that multiple proteins interact with these *PvNAC* genes, forming a complex regulatory network (Figure 6; Appendix A). Notably, PvNAC73 emerged as a central regulator within this PPI network, interacting with key proteins such as BZIP8, DREB2A, and MYC2. DREB2A is a critical transcription factor involved in signal transduction under salt and dehydration stress, and its regulation has been well documented in abiotic stress responses [54]. BZIP8 and MYC2 are known to be key players in stress response, with MYC2 modulating hormonal signaling such as JA/ABA pathways [55], suggesting that PvNAC73 may regulate salt tolerance by modulating the expression of genes involved in stress signaling pathways.

PvNAC06 interacts with proteins such as AP2, CYP72C1, and CLPD, which have roles in transcriptional regulation and stress responses. AP2 has been associated with plant developmental processes and stress tolerance [56], while CYP72C1 and CLPD are linked to hormonal regulation and stress-related metabolic pathways [57,58]. This suggests that PvNAC06 may coordinate stress response and developmental regulation under salt stress. Additionally, PvNAC17 interacts with ribosomal proteins (RPL17B, RPL23AB) and BTF3, which are involved in protein synthesis and transcriptional regulation [59]. These interactions suggest that PvNAC17 may influence salt tolerance by maintaining protein homeostasis and regulating gene expression during stress conditions. PvNAC38 interacts with DREB2A and ZAT10, proteins well known for their roles in abiotic stress responses, particularly drought and salt stress [54,60]. The interaction of PvNAC38 with these proteins suggests that it may contribute to enhancing stress tolerance by regulating genes involved in abiotic stress signaling.

## 3. Materials and Methods

### 3.1. Identification of NAC Gene Family Members in Seashore Paspalum

The whole genome sequences and gene annotation files of seashore paspalum were obtained from the Phytozome database (https://phytozome-next.jgi.doe.gov/, accessed on 25 August 2024), and the full set of NAC protein sequences of *Arabidopsis* thaliana was acquired from the TAIR database (https://www.arabidopsis.org/, accessed on 25 August 2024) for subsequent construction of the evolutionary tree. *NAC* (PF01849) and *NAM* (PF02365) HMM models for *NAC* genes were downloaded from the Pfam database (http://pfam-legacy.xfam.org/, accessed on 25 August 2024) [61]. TBtools(v2.142) software was used to perform HMM analysis on the seashore paspalum protein sequences using the HMMER software package. The hmmsearch command was executed with parameters --cpu 4 -E 1000 to identify potential NAC domain-containing proteins [62]. Following the HMM analysis, the TBtools(v2.142) Venn Graph function was employed to intersect and filter out common genes. Sequences with alignment coverage less than 50% of the HMM model were discarded to ensure the integrity of the NAC domain. After manually removing duplicate sequences, the NCBI Conserved Domain Database (https://www.ncbi.nlm.nih.gov/cdd/?term=, accessed on 25 August 2024) [63] and the Pfam database were used to confirm the presence of the NAC domain in the candidate sequences. The conserved domain analysis was performed with an Expect value cutoff of <0.01, ensuring that only sequences containing the NAC conserved domain were retained for further analysis.

### 3.2. Physicochemical Property Analysis of PvNAC Proteins

The physicochemical properties of the 168 selected *PvNAC* proteins, including molecular weight, amino acid count, instability index, hydrophilicity coefficient, and isoelectric point, were analyzed using the ProtParam tool available through ExPASy (https://web.expasy.org/protscale/, accessed on 20 August 2024). Subcellular localization predictions for these proteins were performed using WoLF PSORT (https://wolfpsort.hgc.jp/, accessed on 25 August 2024).

### 3.3. Phylogenetic Analysis of PvNAC Proteins

Multiple sequence alignment of 168 PvNAC proteins was performed using the built-in Align Protein tool in MEGA 11(v0.13) software, and the results were visualized using Gendoc [64]. A phylogenetic tree was subsequently constructed using the neighbor-joining (NJ) method with the following parameters: Poisson model, pairwise deletion, 1000 bootstrap replicates, and the number of threads set to 3. The phylogenetic tree was visualized as a circular tree using iTOL (https://itol.embl.de/, accessed on 25 August 2024). The *PvNAC* genes were classified into clades based on their topological structures in the circular phylogenetic tree, with only those genes having bootstrap values of 95 or higher being confidently assigned to specific clades. Genes with bootstrap values below 95 were considered as not clearly assignable to any particular group.

### 3.4. Motifs and Gene Structure of PvNAC

The PvNAC protein sequences were uploaded to the NCBI Conserved Domain Database (https://www.ncbi.nlm.nih.gov/cdd/?term=, accessed on 25 August 2024) and the MEME Suite (https://meme-suite.org/meme/, accessed on 25 August 2024) [65] to obtain data of conserved domains and motif patterns. For the MEME analysis, the parameters were configured as follows: the background model was adjusted to account for single-letter biases (0-order model), the motif width was set with a minimum of 6 and a maximum of 50, and the number of motifs to be identified was set to 15. The data were subsequently visualized using the Gene Structure View (Advanced) tool in TBtools(v2.142), which offered a detailed and comprehensive overview of the gene structures and conserved domains within the PvNAC protein family.

### 3.5. Cis-Element Analysis of the PvNAC Gene Family

The complete gene set and the genome annotation file of seashore paspalum were submitted to TBtools(v2.142). The promoter regions, consisting of 2000 bp upstream of the start codon of the *PvNAC* genes, were extracted using the TBtools(v2.142)-GXF Sequences Extract tool. The sequences were then submitted to the PlantCARE database (https://bioinformatics.psb.ugent.be/webtools/plantcare/html/, accessed on 26 August 2024) [66] for the analysis of cis-acting regulatory elements. The resulting tabulated data were subsequently visualized using the TBtools(v2.142)-Basic Biosequence View tool.

### 3.6. Collinearity and Evolution Analysis of PvNACs

Based on the identified 168 *PvNAC* sequences and the genome annotation file of seashore paspalum, collinearity analysis was performed using the TBtools(v2.142)-One Step McscanX tool with the following parameters: number of BLAST hits set to 5 and E-value set to 1 × 10^−10^ The collinearity analysis results were subsequently visualized using the TBtools(v2.142)-Advanced Circos tool. 

### 3.7. Functional Annotation Analysis of PvNAC Proteins

Gene ontology (GO) annotation was performed by uploading all PvNAC protein sequences to the eggNOG database (http://eggnog-mapper.embl.de/, accessed on 15 October 2024) [67]. The returned results were then used for GO enrichment analysis and visualization using TBtools(v2.142) software (version 2.127) [68].

### 3.8. Co-Expression Network Construction

The protein interaction network was constructed using STRING (https://cn.string-db.org/cgi/input?sessionId=bsYB1aGF0p8U&input_page_active_form=annot_proteome, accessed on 26 August 2024) with *Arabidopsis thaliana* as the reference species under medium confidence (0.400) and the network type set to the full STRING network with an FDR stringency of medium (5 percent). The resulting data were subsequently visualized using Cytoscape (version 3.10.2). In the visualization, protein size and color were mapped continuously according to the degree value between proteins.

### 3.9. Plant Growth and Treatments

Single stem nodes of seashore paspalum were transplanted to 6 hydroponic boxes (30 cm × 30 cm) and cultured with Hoagland’s nutrient solution under identical greenhouse conditions for 12 weeks. Throughout the experiment, the temperature was maintained at 30/25 °C (day/night), and the relative humidity was kept at 50% during the day and 70% at night. The plants exhibited healthy growth and thrived under these conditions.

The control group was maintained with Hoagland’s nutrient solution and labeled as the 0 h treatment. Salt treatment was conducted by adding 0.2 M NaCl to Hoagland’s nutrient solution. Leaf and root samples were rapidly collected at 6, 48, and 120 h after the initiation of salt treatment. During salt treatment, the leaves initially became soft but remained spread at 6 h. Symptoms of dehydration and curling appeared at 48 h. As the treatment duration extended to 120 h, the curling became more pronounced, and the basal leaves began to yellow and wither. Samples were taken from the second or third fully developed leaf from the top. The samples were immediately flash-frozen in liquid nitrogen and, respectively, stored at −80 °C.

### 3.10. RNA Extraction and Illumina Sequencing

Total RNA was extracted from leaves and roots at four time points (0, 6, 48, and 120 h) from three independent biological replicates per treatment, using the Plant Total RNA Extraction Kit (TIANGEN Biotech, Beijing, China), following the manufacturer’s protocol. cDNA was synthesized using the PrimeScript RT reagent kit with a gDNA eraser (Takara, Dalian, China). The quality and integrity of the RNA samples were evaluated using the Agilent 2100 Bioanalyzer (Agilent Technologies, Santa Clara, CA, USA), NanoDrop2000 (Thermo Fisher Scientific, Waltham, MA, USA), and agarose gel electrophoresis. Sequencing libraries were prepared with the NEBNext Ultra™ RNA Library Prep Kit for Illumina (NEB, Ipswich, MA, USA) according to the manufacturer’s guidelines. For RNA-seq analysis, eight libraries were constructed: four from leaf samples (L-0, L-6, L-48, and L-120) and four from root samples (R-0, R-6, R-48, and R-120), each with three biological replicates, resulting in a total of 24 libraries for leaves and 24 libraries for roots. The sequencing was performed using the Illumina NovaSeq 6000 system (Illumina, San Diego, CA, USA) by Biomarker Technologies (Beijing, China), and the RNA-seq data were generated using paired-end sequencing. Quantification of gene expression levels was estimated by fragments per kilobase of transcript per million (FPKM) mapped reads. A total of 42,893,496 clean reads were generated for sample L0_1, with similar numbers for other samples. The reads were aligned to the reference genome using the HISAT2 algorithm [69]. Each cDNA library required ≥1 µg of total RNA, an OD260/280 ratio of ≥1.8, and a RIN value of ≥6.5. RNA-seq analysis proceeded after confirming sample quality.

The fold changes (FC) between the control (0 h) and salt treatments (6, 48, and 120 h) were calculated based on the FPKM values from the transcriptome data. The fold changes at each time point relative to 0 h were normalized using log_2_(fold change). Heat maps were generated using Tbtools(v2.142) software to visualize the differentially regulated genes across the various treatment time points.

### 3.11. Quantitative Real-Time PCR Analysis

Specific primers for qRT-PCR were designed using Primer Premier 5.0 software with melting temperatures between 55–65 °C, primer lengths between 19–22 bp, and amplification lengths between 100–300 bp. All primers used for qRT-PCR were listed in Appendix A. The qRT-PCR reaction volume was 20 μL, consisting of 10 μL of 2× concentrated SYBR Green I premix, 5 μL of cDNA, 1 μL of each primer, and 3 μL of double-distilled water. The reaction protocol was as follows: initial denaturation at 95 °C for 10 min, followed by 45 cycles of denaturation at 95 °C for 15 s, annealing at 58 °C for 15 s, and melt curve generation from 60 to 95 °C. Each qRT-PCR experiment was performed with three biological replicates, and each biological replicate was measured in three technical replicates. The data were normalized using the 2^−ΔΔCT^ method [70]. *PvActin1* [34] and *PvSAND* (KX268093) [71], the most stable housekeeping genes during salt treatment, were selected as internal housekeeping genes. Statistical analyses were performed using one-way ANOVA followed by Duncan’s multiple range test (*p* < 0.05). Data are presented as mean ± standard error (SE) of three biological replicates.

### 3.12. Statistical Analysis

All experiments were repeated 3 times, utilizing 3 independent RNA samples to ensure the reliability and reproducibility of the results. Data from 3 replicates were analyzed by using one-way ANOVA. All statistical analysis was performed by the Statistical Package for the Social Sciences (SPSS 17.0). Results are shown as mean ± standard error of biological replications. The means were separated using Duncan’s multiple range test (*p* < 0.05) [72].

## 4. Discussion

The *NACs* in seashore paspalum, a significant member of the plant-specific transcription factor superfamily, play an essential role in various aspects of plant growth and development. NAC proteins are multifunctional, contributing critically to the plant’s response to both biotic and abiotic stresses [73]. This includes responses to salt stress [14], high-temperature stress [48], water stress [74], and oxidative stress [75], among other abiotic challenges. Additionally, they regulate functions such as seed dormancy [76], cellular senescence [77], and cell wall growth [78], demonstrating their versatile regulatory capabilities. Seashore paspalum, also known as seaside paspalum, is a halophyte with significant ecological and economic value, and its adaptability to salt stress is of great importance for study. However, there is currently a relative scarcity of research data on the *NAC* gene family in this specific species. This study aims to conduct a comprehensive genomic analysis of the *NAC* gene family in seashore paspalum to reveal its potential functions in response to salt stress. Through systematic identification and functional analysis of the seashore paspalum *NAC* gene family, we can deduce information about the *PvNAC* genes, such as chromosomal location, gene structure, and gene collinearity relationships. This will help us understand the role of these genes in the plant’s adaptation to stress and provide a theoretical foundation and molecular biological basis for the genetic improvement and breeding of seashore paspalum.

Using whole-genome data of seashore paspalum, we identified 168 *PvNAC* genes through a hidden Markov model of the conserved *NAC* (*NAM*) domain. The physicochemical properties of PvNAC proteins revealed that all exhibit a negative hydrophilicity coefficient, indicating hydrophilicity. Additionally, 26 proteins (15.48%) had an instability index exceeding 40, suggesting most are unstable, while the isoelectric point (pI) ranged from 3.17 to 10.40, allowing functionality across various pH conditions. Phylogenetic analysis using the neighbor-joining (NJ) method grouped the 168 *PvNAC* genes into 18 subgroups, with members within each group exhibiting strong structural similarity. This classification aligns with the conserved structure observed in *NAC* gene families across various plant species, such as *Nelumbo nucifera*, *Arachis duranensis*, and *Arachis ipaensis.* For instance, 97 *NAC* genes identified in *N. nucifera* were similarly divided into 18 subgroups based on phylogenetic analysis [79], and 81 and 79 *NAC* genes were classified into 18 subgroups in *A. duranensis* and *A. ipaensis*, respectively [80]. After constructing a phylogenetic tree of PvNAC proteins alongside Arabidopsis NAC proteins (Appendix A), we found that PvNAC52 clustered closely with Arabidopsis AtNAC16. In Arabidopsis, *AtNAC16* (*AT1G73830*) plays a crucial role in the drought stress response. Studies have shown that the NAC016 mutant exhibits enhanced drought tolerance, while plants overexpressing NAC016 show reduced drought tolerance [81]. Additionally, we found that PvNAC86, PvNAC114, and PvNAC151 formed a tight cluster with *Arabidopsis* AtNAC06 (ATAF1) and AtNAC26. *AtNAC06* (*ATAF1*) is known to play a key role in the ABA signaling pathway and salt stress tolerance by regulating reactive oxygen species (ROS) scavenging mechanisms, thereby enhancing plant resistance to salt stress. This close evolutionary relationship suggests that PvNAC86, PvNAC114, and PvNAC151 may share similar regulatory roles in promoting salt tolerance in Seashore Paspalum, likely through mechanisms involving ABA signaling and ROS scavenging [48]. This suggests that the close evolutionary relationship between PvNAC52 and AtNAC16, as well as between PvNAC86, PvNAC114, PvNAC151, and AtNAC06 (ATAF1), indicates these PvNAC proteins may function in analogous pathways to enhance stress resilience in Seashore Paspalum. These findings suggest that the grouping of *PvNAC* genes is highly conserved within the NAC family, supporting their functional conservation across different plant species. Promoter analysis of *PvNAC* genes revealed a high abundance of light-responsive cis-acting elements, which were present in nearly all members, emphasizing the importance of light in regulating *PvNAC* gene expression. Hormone-responsive elements, including abscisic acid (ABA), gibberellin, and MYB binding sites, were also widely distributed, suggesting a role in integrating environmental and hormonal signals. These cis-elements are consistent with findings in other species; for example, *A. duranensis* and *A. ipaensis* showed that *NAC* gene promoters contain key elements responsive to light, hormones, and abiotic stresses. The distribution of these elements underscores the multifunctionality of *NAC* genes, particularly their involvement in transcriptional regulation during stress responses and growth processes. We also found that the promoter regions of genes such as *PvNAC28*, *PvNAC29*, *PvNAC30*, *PvNAC31*, *PvNAC32*, *PvNAC33*, *PvNAC34*, *PvNAC35*, etc., show a high degree of similarity in cis-acting elements, which is speculated to be related to the functional relevance of this gene cluster. GO annotation and enrichment analysis revealed that *PvNAC* genes play crucial roles in transcriptional regulation, organ development, and responses to environmental stimuli. They are primarily involved in metabolic processes and transcription factor activity and are predominantly localized in the nucleus and membrane-bound organelles. This highlights the essential functions of *PvNAC* genes in regulating gene expression and plant stress responses. Gene duplication events, inferred from collinearity analysis, suggest that the *PvNAC* gene family likely underwent gene duplication, which may have contributed to its expansion and functional diversification, enabling seashore paspalum to adapt to saline environments.Gene duplication is a well-established evolutionary mechanism that allows plants to acquire novel functions or enhance existing ones [82]. The high similarity observed in promoter regions among certain *PvNAC* genes, such as *PvNAC28* to *PvNAC35*, may indicate coordinated regulation and overlapping functions, which are likely advantageous under stress conditions. Similarly, in *Pyrus bretschneideri*, 183 *NAC* genes (*PbNACs*) were identified, which were found to cluster in chromosomal fragments rather than being evenly distributed—likely due to uneven duplication events in the pear genome [83]. In red clover (*Trifolium pratense*), 72 TpNAC genes were identified, with collinearity analysis revealing five large segmental duplication pairs [84].

To explore the *PvNAC* genes involved in regulating the salt tolerance response, we conducted salt stress experiments, sampled the root system and leaves, and measured the expression levels of *PvNACs*. According to the heat map of *PvNAC* expression levels, under treatment with 0.2 mol/L NaCl, most *PvNAC* genes responded, with the response level in the root system generally being greater than that in the leaves. This organ-specific expression pattern suggests that these genes may play critical roles in root-mediated salt tolerance mechanisms, such as ion homeostasis, osmotic adjustment, and root architecture modification. For instance, in *Arabidopsis*, the *NAC* gene *AtNAC042* (also known as RD26) is induced by salt stress and regulates the expression of stress-responsive genes involved in detoxification and antioxidant processes [85]. Subsequently, we selected 7 *PvNAC* genes with higher expression levels (*PvNAC06*, *PvNAC17*, *PvNAC38*, *PvNAC53*, *PvNAC71*, *PvNAC73*, *PvNAC142*, *PvNAC162*) for qRT-PCR verification. The results showed that the expression trends of qRT-PCR were consistent with the expression trends of RNA-seq data. To better understand the roles of *PvNAC* genes, we constructed a protein interaction network for *PvNAC06*, *PvNAC17*, *PvNAC38*, and *PvNAC73* based on the direct orthologous species *Arabidopsis* thaliana. The results showed that the above four PvNAC proteins formed a complex protein interaction network with 33 other proteins, with *PvNAC73* predicted to interact with *PvNAC06*, *PvNAC17*, and *PvNAC38*, and proteins such as *BZIP8* and *DREB2A*, which are considered to be involved in stress signal transduction, forming a central regulatory chain. Moreover, the expression profiles of *PvNAC* genes observed in our study align with those reported in other salt-tolerant species, supporting the conserved role of *NAC* genes in plant stress tolerance. For instance, in *Miscanthus sinensis*, the SNAC subgroup, which shares structural similarities with the *PvNAC* genes identified here, showed significant upregulation under salt stress, implicating these genes in the modulation of salt tolerance mechanisms [86]. Similarly, in *Dendrobium nobile*, the expression of *DnoNAC* genes was shown to be strongly responsive to varying salt concentrations, underscoring the salt sensitivity of *NAC* genes across different species [87]. A comparison with salt stress-responsive *NAC* genes in other species further emphasizes the broad functional significance of this gene family. For example, in *Tritipyrum*, 68 *TtNAC* genes were highly expressed in response to salt stress, and one of them, *TtNAC477*, was specifically upregulated in roots, stems, and leaves, suggesting its potential utility in improving salt tolerance in wheat [88]. Similarly, in *Medicago sativa*, 47 salt-responsive *MsNAC* genes were identified, with 35 showing upregulation under salt stress, which were validated through RT-qPCR, further validating the role of *NAC* genes in salt adaptation mechanisms [89]. The findings from this study are consistent with reports on halophytes, such as *Suaeda liaotungensis*, where the *SlNAC10* gene enhances salt and drought tolerance by regulating proline synthesis through transcriptional regulation of key biosynthetic genes like *AtP5CS1* and *AtP5CS2* in *Arabidopsis* [90]. This illustrates a conserved mechanism whereby *NAC* genes regulate osmotic stress responses, such as proline accumulation, across species. Additionally, the role of NAC proteins in ABA signaling, as evidenced by the function of *PagNAC045* from *Populus alba* L. in transgenic tobacco, highlights the potential of *NAC* genes to modulate both salt and ABA responses, further supporting their role in stress signal transduction [91].

In future studies, a more detailed exploration of the gene function and regulatory pathways underlying the salt tolerance response will be essential. Specifically, investigating the downstream target genes regulated by PvNAC proteins and their interactions with other key signaling pathways, such as those involving ABA, ROS, and ion transporters, will offer deeper insights into the molecular mechanisms of plant stress responses. Such studies will help elucidate how PvNAC proteins coordinate complex stress responses at the transcriptional level, influencing not only salt tolerance but also other forms of abiotic stress. Understanding the crosstalk between NAC proteins and other stress-related signaling pathways will be crucial for designing strategies to enhance salt tolerance in crops through genetic modification. Furthermore, functional validation of *PvNAC* gene candidates in model organisms and crop species will provide invaluable information on their potential as targets for improving stress resilience in agriculture. By integrating transcriptomic, proteomic, and metabolomic data, future research will be able to create a more holistic view of how NAC proteins regulate plant responses to environmental stressors, paving the way for the development of salt-tolerant crops with improved yield stability under stress conditions.

## 5. Conclusions

This study provides a comprehensive genome-wide analysis of the *NAC* gene family in seashore paspalum (*Paspalum vaginatum*), identifying 168 *PvNAC* genes across 18 subgroups. Structural analysis revealed conserved motifs, domain architectures, and cis-regulatory elements, highlighting their complex regulatory potential. Gene duplication events were also observed, suggesting an evolutionary adaptation to environmental stress.RNA-seq and qRT-PCR revealed eight *PvNAC* genes significantly upregulated under salt stress, particularly in roots, suggesting their role in salt tolerance through mechanisms like ion homeostasis, osmotic balance, and antioxidant defense. Protein interaction analysis identified PvNAC73 as a key regulator, interacting with stress-responsive proteins such as BZIP8 and DREB2A. Future research should focus on the functional validation of key *PvNAC* genes to elucidate their precise roles in salt tolerance mechanisms. This should include investigating their regulatory interactions with other transcription factors, signaling pathways, and environmental stress-related genes, as well as identifying their downstream targets involved in ion homeostasis, osmotic regulation, and antioxidant defense. Additionally, cross-species comparisons, especially between salt-tolerant and sensitive species, will provide deeper insights into the evolution of *NAC* gene functions. Ultimately, these efforts could lead to the development of salt-tolerant crops, contributing to more sustainable agricultural practices in saline and coastal environments and addressing global challenges posed by soil salinity.

## Figures and Tables

**Figure 1 plants-13-03595-f001:**
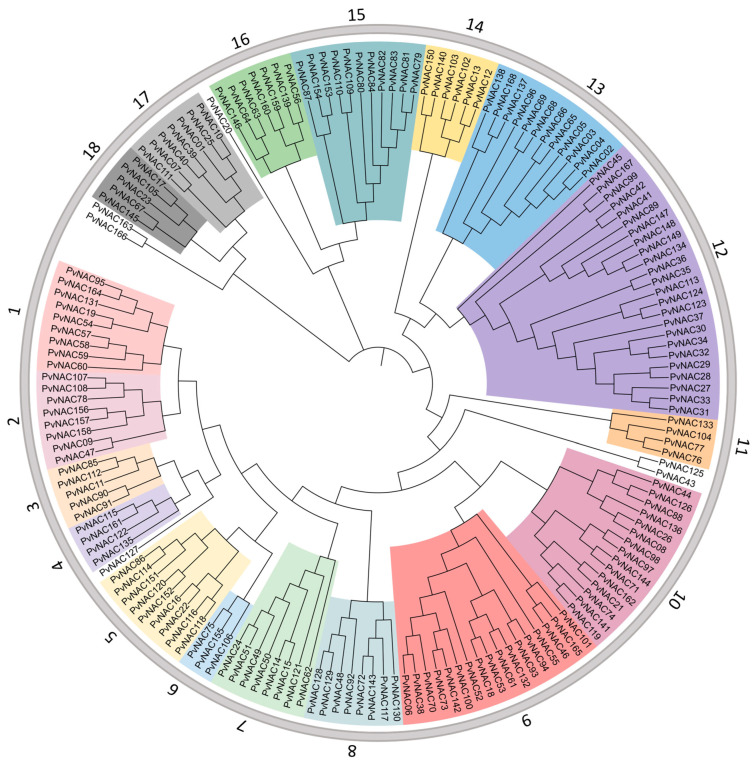
Phylogenetic tree of the *PvNAC* gene family in seashore paspalum. Different colors represent distinct subgroups, with clustering indicating potential functional similarities among closely related genes within each clade.

**Figure 2 plants-13-03595-f002:**
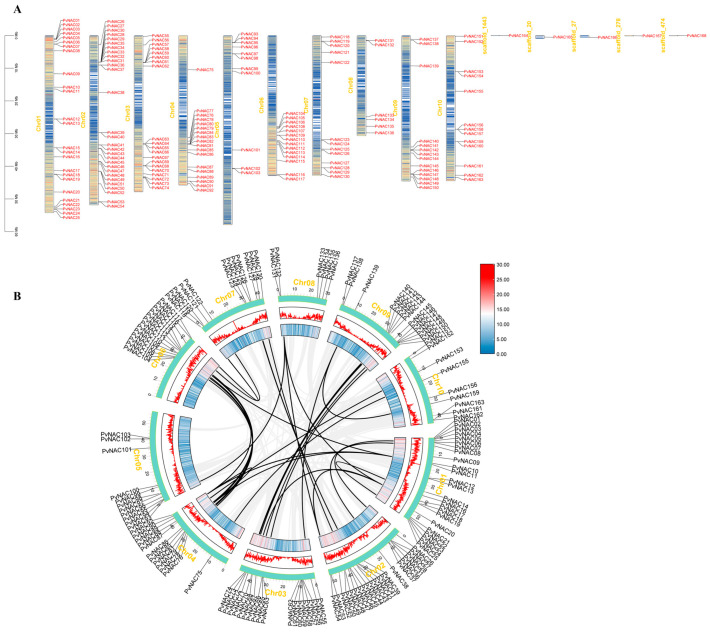
Chromosomal localization and syntenic relationships of *PvNAC* genes. The chromosomal distribution of the 168 *PvNAC* genes within the seashore paspalum genome is depicted. Each *PvNAC* gene is labeled according to its chromosomal location (**A**). The syntenic relationships are illustrated, with large-scale duplications indicated by orange lines connecting duplicated *NAC* gene pairs. The gray regions represent syntenic blocks, while the orange lines highlight the duplicated gene pairs, suggesting potential evolutionary events within the *NAC* gene family (**B**).

**Figure 3 plants-13-03595-f003:**
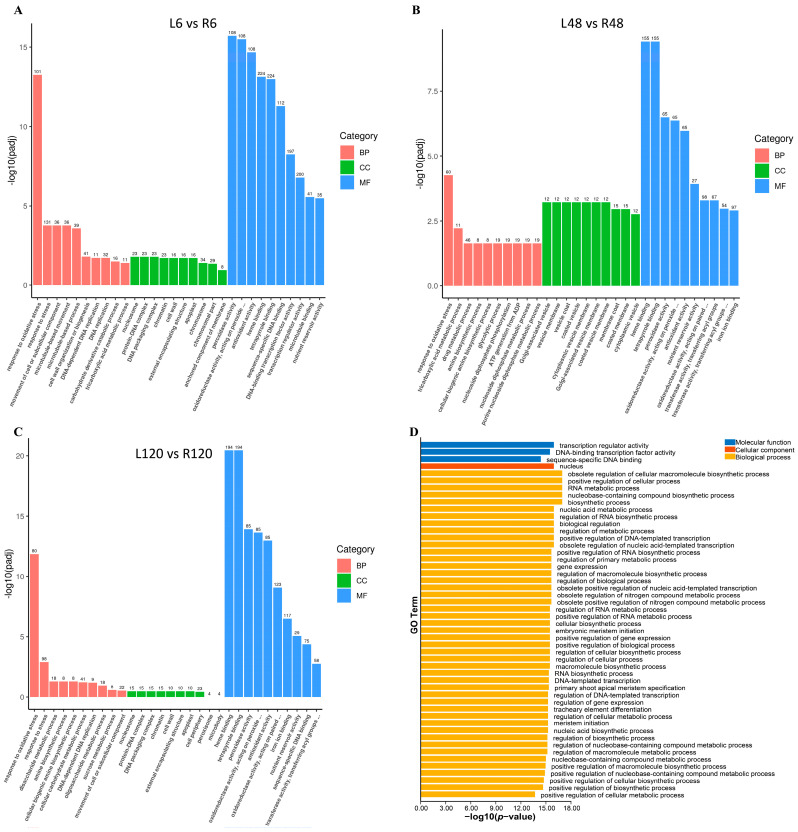
Gene ontology (GO) enrichment analysis of differentially expressed genes (DGEs) in seashore paspalum. Comparative DEGs from L6 vs. R6 (**A**), L48 vs. R48 (**B**), and L120 vs. R120 (**C**) were analyzed via GO, with enriched terms in biological process, cellular component, and molecular function displayed as a histogram. The X-axis displayed the −log_10_(*p*-value), reflecting the statistical significance of the enriched GO terms of PvNAC, with higher values indicating greater significance of enrichment (**D**).

**Figure 4 plants-13-03595-f004:**
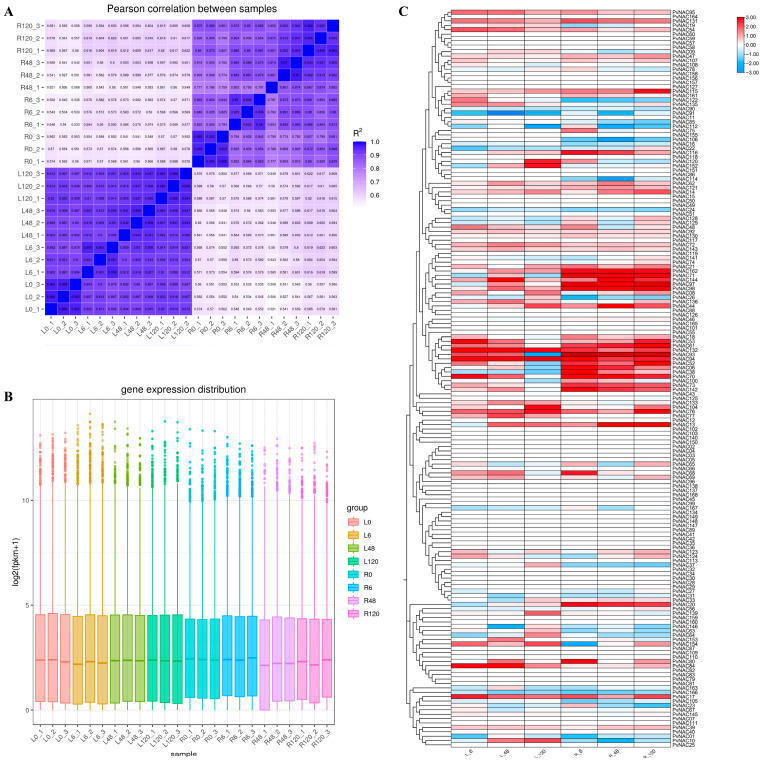
Comprehensive Analysis of *PvNAC* Gene Expression Under 0.2 M NaCl Treatment. (**A**) This heatmap depicts the Pearson correlation coefficients (R^2^) between all experimental samples. Both the X and Y axes list the sample identifiers, while the color intensity represents the strength of the correlation, with a gradient from blue (low correlation) to dark red (high correlation). Darker shades indicate stronger positive correlations (R^2^ approaching 1), facilitating the assessment of sample similarity and reproducibility. (**B**) The box plot illustrates the distribution of normalized *PvNAC* gene expression levels (log_2_(FPKM + 1)) across all samples. The X-axis categorizes the samples by their respective names, and the Y-axis quantifies the expression levels. Each box represents the interquartile range (IQR) with the median indicated by the horizontal line, while whiskers extend to the minimum and maximum values, excluding outliers. (**C**) This heatmap showcases the differential expression of *PvNAC* genes in leaf (L) and root (R) tissues at three distinct time points (6, 48, and 120 h) under 0.2 M NaCl treatment. Sample identifiers are denoted as L-6, L-48, and L-120 for leaves and R-6, R-48, and R-120 for roots. Expression levels are color-coded using a red-to-blue gradient, where red signifies upregulation and blue indicates downregulation relative to the 0 h control.

**Figure 5 plants-13-03595-f005:**
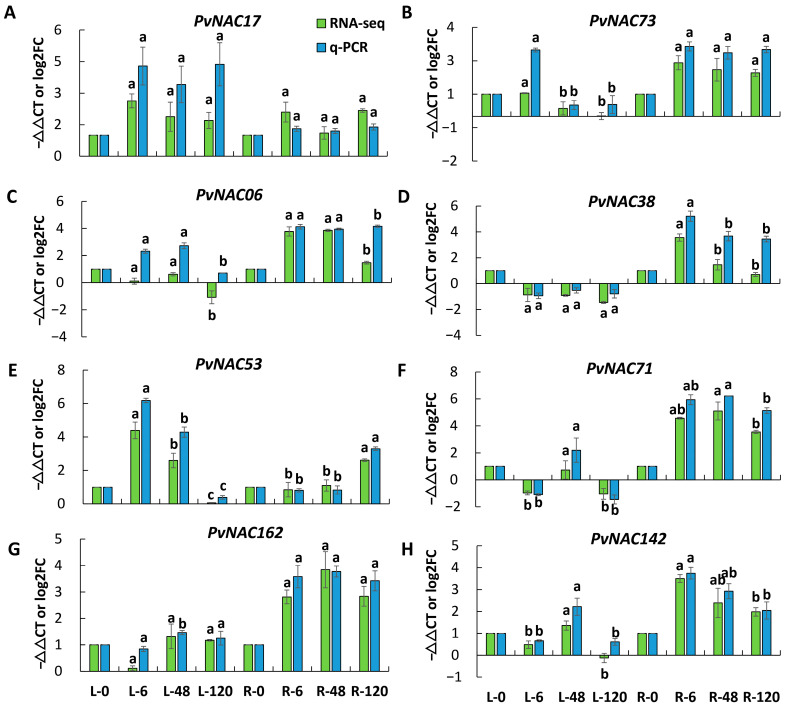
Expression levels of *PvNAC* genes in roots and leaves under salt stress. (**A**–**H**) illustrate the trends of transcriptome and gene expression levels of NAC genes under salt stress**.** Transcriptome data were presented as Log2 fold changes of FPKM values, comparing treatments L-6, L-48, L-120, R-6, R-48, and R-120 against the control groups L-0 and R-0, respectively. The relative gene expression levels, as determined by qRT-PCR, were quantified using the Log_2_(2^−ΔΔCt^) method, further validating the trends in gene expression changes. Different letters above the bars indicated statistically significant differences determined by ANOVA *(p* < 0.05). The presented data represent the means of three independent experiments, with error bars denoting the standard error.

**Figure 6 plants-13-03595-f006:**
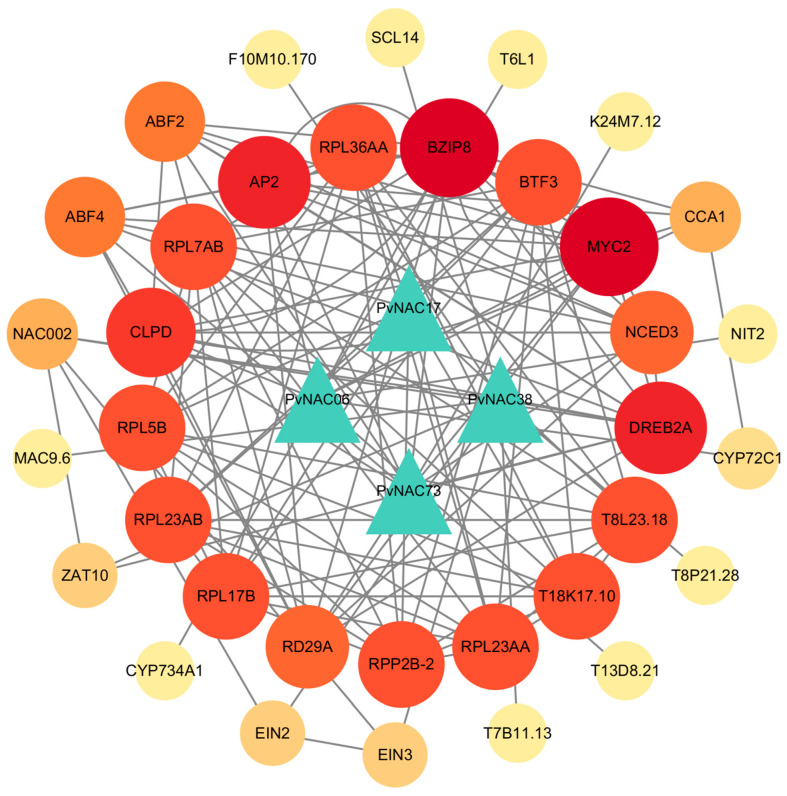
Protein–protein interaction network of PvNAC proteins. The network was constructed with a medium confidence threshold of 0.4. Node color intensity corresponds to the degree value, with darker shades indicating higher connectivity (more interactions). Triangles represent PvNAC proteins (the target proteins in this study), while circles denote their interacting partners. Solid gray lines between nodes indicate protein–protein interactions, with uniform line density and connection strength across the network.

## Data Availability

The datasets generated and analyzed during the current study are not publicly available given the restrictions to data sharing imposed, but de-identified data are available from the corresponding author on reasonable request.

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
