# Peer review of "Genome-Wide Identification and Expression Analysis of NAC Gene Family Members in Seashore Paspalum Under Salt Stress"

_plants, 2024, doi:10.3390/plants13243595_

Round 1

Reviewer 1 Report (Previous Reviewer 2)

Comments and Suggestions for Authors

Thanks for the response to our previous comments. We understand that much effort was taken to address the points raised. However, I still have some concerns.

While it is obvious that the MS includes 8 figures, more figures do not necessarily enhance the clarity or impact of the work. I recommend placing some of the less critical, descriptive results in the supplementary materials, such as Fig. 2 and Fig. 3 (actually they were not plotted), as these mainly provide descriptive information that does not directly relate to the central theme of the study on NAC Gene Family Members in Seashore Paspalum under Salt Stress. Meanwhile, Fig. 1, Fig. 6, and Fig. 7 present relevant results that contribute meaningfully to the main narrative and improve the manuscript significantly.

Additionally, Table S1 and Table S6 still contain non-English characters. I am unsure if this is due to an incorrect file upload or if certain sheets have been overlooked.

Comments on the Quality of English Language

None

Author Response

尊敬的评论者: 我们衷心感谢您的周到和耐心的建议,这些建议为提高我们的稿件质量做出了巨大贡献。 请在附件中找到我们的回答。 我们再次非常感谢您为改进我们的工作提供的帮助。祝您生活愉快,工作圆满。

Reviewer 2 Report (New Reviewer)

Comments and Suggestions for Authors

The manuscript addresses a critical topic—understanding the role of NAC transcription factors in seashore paspalum’s response to salt stress. While the research presents considerable bioinformatics analysis and experimental validation, several aspects need significant improvement to ensure scientific rigor and enhance the manuscript’s overall quality. Below is a detailed critique and recommendations for improvement:

Major Concerns and Recommendations:

·       The introduction provides a strong background on NAC transcription factors, it lacks a clear articulation of the study’s novelty. Explicitly state the knowledge gap addressed by this research.

·       Expand on the evolutionary relationships among NAC genes, discussing their functional implications. Provide examples of orthologous NAC genes in model plants and their roles in abiotic stress responses.

·       The methodology for identifying NAC genes using HMMER is not clearly outlined. Specify the e-value thresholds, alignment coverage, or domain-specific filters applied during gene selection. Describe any additional filtering criteria (e.g., removing redundant sequences or confirming conserved domains with Pfam or SMART tools).

·       Link the identified promoter motifs to specific NAC functions under salt stress. For example, discuss how specific motifs (e.g., ABRE, MYB) influence transcriptional regulation and provide functional hypotheses.

·       Include detailed information about experimental replicates (biological and technical) and statistical analysis methods used to ensure data reliability. Mention the number of replicates, normalization techniques, and the statistical significance thresholds applied.

·       Improve the quality and clarity of key figures, especially Figures 6 and 8. Add detailed annotations, color coding, or legends to make data interpretation more accessible. Ensure consistency in formatting across all figures.

·       Address grammatical errors and revise unclear statements to improve comprehension. Streamline sections with redundancy, particularly in the phylogenetic and cis-element analysis discussions.

·       The discussion section needs a stronger focus on mechanisms. Avoid reiterating results and instead explore the implications of key findings. Compare your results with studies on NAC genes in other plant species to contextualize them.

·       Discuss the potential utility of PvNAC genes in molecular breeding for salt tolerance. Provide specific examples of how these genes could be targeted for improving stress resilience in crops.

·       Revise the conclusions section to focus on the most critical findings and their broader implications. Highlight specific areas for future research, such as functional characterization of key PvNAC genes or exploring their interactions with other regulatory elements.

Comments on the Quality of English Language

Please improve the language quality

Author Response

Dear Reviewer,     We sincerely thank you for your thoughtful and patient suggestions, which have greatly contributed to improving the quality of our manuscript. Please find our responses in the attached document. Once again, we deeply appreciate your help in enhancing our work. Wishing you a pleasant life and success in your work.

Round 2

Reviewer 1 Report (Previous Reviewer 2)

Comments and Suggestions for Authors

The authors have almost addressed all of my concerns. I hope the MS will be further refined in detail during the proofreading stage.

Reviewer 2 Report (New Reviewer)

Comments and Suggestions for Authors

the authors addressed all comments and improved the manuscript  accordingly. i suggest to proceed it positively. 

This manuscript is a resubmission of an earlier submission. The following is a list of the peer review reports and author responses from that submission.

Round 1

Reviewer 1 Report

Comments and Suggestions for Authors

The authors have thoroughly revised the manuscript and addressed all my comments and suggestions. The methods section in particular has been significantly improved. 

Reviewer 2 Report

Comments and Suggestions for Authors

This manuscript provides a good analysis of NAC transcription factors and their roles in plant stress responses. The study addresses interesting questions regarding the contribution of these transcription factors to stress tolerance in plants. The authors have employed a variety of experimental approaches, including gene expression analysis, functional studies, and phenotypic assessments, to present a holistic view of NAC transcription factors' functions. While the results are generally interesting and relevant, I have several concerns that need to be addressed.

Major Points:

  1. Figure legends: The figure legends are quite brief and do not comprehensively cover all the information presented in the corresponding figures. Providing more detailed explanations in the legends would help readers better understand the data.
  2. For Figure. 1, since it represents a single-gene tree, I suggest also including the results of a multi-gene alignment. Through the multi-gene alignment, it becomes evident which regions are conserved and under strong selected. Please refer to Fig. S2 (https://doi.org/10.1016/j.virusres.2020.198145) and consider adding it as a supplementary figure. Additionally, in Fig. 1, to enhance the visual appeal of the tree, I recommend using iTOL to color each branch refer to (https://doi.org/10.1016/j.cell.2018.10.023 https://doi.org/10.1016/j.cell.2022.06.014). Displaying the corresponding bootstrap values on the branches will further indicate the reliability of the constructed phylogenetic tree.
  3. Figure quality and organization: The figures are not very well organized. For example, Figure 2 shows information on the PvNAC gene motifs, but it seems that some motifs have not been fully identified, or the information provided is insufficient. Additionally, the image quality appears unclear. I recommend uploading all figures in SVG or PDF format for improved clarity. Or this part is not import and could be transferred into a supplement figure. The same applies to Figure 3. In Figure 4, the Circos plot appears compressed into an oval shape.
  4. Expression levels of PvNAC genes: There are some gene expression levels reported as negative values, which is confusing. It is unclear what these negative values represent. A clearer explanation of this is needed to help readers interpret the results or there exists some error.
  5. Supplementary Tables: Tables S1, S3, and S5 appear to contain non-English (possibly Chinese) text.
  6. Supplementary materials: I suggest adding additional supplementary files, such as the phylogenetic tree in Newick format, including bootstrap values. This would allow readers to better understand and verify the phylogenetic results presented.
  7. Some figures, such as Figure 7 showing the PPI network, seem more descriptive and lack strong evidence to support their relevance or contribution to the main narrative. For example, does the author provide direct experimental validation of the interactions, such as through co-immunoprecipitation (CoIP) or dual-luciferase assays? If these interactions are not extensively validated or do not add significant value to the overall structure, I suggest considering moving these figures to the supplementary section.

Comments on the Quality of English Language

Mino points,

I noticed some overlap between the discussion and results sections. I recommend refining and condensing these sections to avoid repetition and enhance the clarity of your manuscript.

The discussion section should elevate the manuscript beyond simply reporting results. It provides a deeper exploration of the study's significance, connects findings to previous research, identifies current gaps, and suggests future research directions. 

Reviewer 3 Report

Comments and Suggestions for Authors

The original work proposed here is an extensive analysis of NAC gene family members in Seashore Paspalum. While the title suggests a focus on salt stress conditions, I do not see this point substantially addressed in the work. The sequence analysis is primarily bioinformatic, and I miss more molecular evidence to support these findings. For instance, microscopy assays showing the localization of representative NAC proteins, yeast two-hybrid assays to study predicted interactions from in silico analysis, or qRT-PCR data demonstrating changes in gene expression to validate the RNA-seq results would greatly enhance the study.

Additionally, why are the authors normalizing the RNA-seq data using FPKM? It is widely accepted that this normalization underestimates mRNA length, and the authors also show significant variability in gene length within the NAC family. In my opinion, the data should be reanalyzed using RPKM normalization. Moreover, the Materials and Methods section requires more detail on this analysis: Was the RNA-seq single-end or paired-end? Which algorithm was used for data analysis? How many reads were generated? A Gene Ontology (GO) study is also missing.

Overall, while this work makes extensive use of bioinformatic tools, it needs improvement in molecular assays to confirm the predictions. For this reason, I would suggest rejection and encourage resubmission after these data are strengthened.

Comments on the Quality of English Language

I would suggest rejection and encourage resubmission after these data are strengthened.